# Exploring tissue morphodynamics using the photoconvertible Kaede protein in amphioxus embryos

**Lydvina Meister** [ORCID]**, Hector Escriva***, **Stéphanie Bertrand** [ORCID]*

Sorbonne Université, CNRS, Biologie Intégrative des Organismes Marins, BIOM, Banyuls-sur-Mer, France

* stephanie.bertrand@obs-banyuls.fr (SB); hescriva@obs-banyuls.fr (HE)

**Data Availability Statement:** All relevant data are within the paper and its Supporting information files.

**Funding:** HE recieved financial support from the "Agence Nationale de la Recherche" under the grants ANR-19-CE13-0011-01 and ANR-16-CE12-

## Abstract

Photoconvertible proteins are powerful tools widely used in cellular biology to study cell dynamics and organelles. Over the past decade, photoconvertible proteins have also been used for developmental biology applications to analyze cell lineage and cell fate during embryonic development. One of these photoconvertible proteins called Kaede, from the stony coral *Trachyphyllia geoffroyi*, undergoes irreversible photoconversion from green to red fluorescence when illuminated with UV light. Undertaking a cell tracing approach using photoconvertible proteins can be challenging when using unconventional animal models. In this protocol, we describe the use of Kaede to track specific cells during embryogenesis of the cephalochordate *Branchiostoma lanceolatum*. This protocol can be adapted to other unconventional models, especially marine animals.

## Introduction

One of the main questions in developmental biology is to understand the different behaviors adopted by the plethora of cells that are generated during embryogenesis through successive cell divisions. How, from a unique cell, a multicellular organism with a specific shape and cellular organisation is built up? Cell lineage tracing analysis refers to the study of cellular genealogies by following cell divisions and migration over developmental time. Such experiments rely on non-invasive labelling techniques that allow tracking cell shape and migration. Vital stains were the first tools to be used for such purposes, but the discovery of fluorescent proteins and of the photoactivation and photoconversion properties of some of them, as well as the development of confocal laser microscopy allowing 3D imaging of fluorescently labelled cells, has brought new opportunities to undertake fine cell-tracing analyses. The protein called Kaede, from the stony open brain coral *Trachyphyllia geoffroyi*, was one of the first photoconvertible molecule specifically used for cell tracking [1]. This protein emits green fluorescence when excited with blue light. After photoconversion using a pulse of UV irradiation, the protein switches irreversibly its emission from green to red. By photoconverting Kaede in a subset of cells they can be easily followed through development *via* time-laps imaging or by imaging the embryo at different developmental stages. Such an approach has been used for example to track cells from the hindbrain of the zebrafish embryo [2] or to

0008-01 and from the European project Assemble Plus (H2020-INFRAIA-1-2016- 2017; grant no. 730984). The funders had no role in study design, data collection and analysis, decision to publish, or preparation of the manuscript.

**Competing interests:** The authors have declared that no competing interests exist.

study the development of lymphoid organs in mammals [3]. Although many cell-tracing experiments using Kaede are described in the literature for classical animal models [4–7], few data are available on the use of this tool in non-vertebrate marine species. Kaede was recently used in the sea urchin early embryo to distinguish pre-existing proteins from newly synthesized/imported proteins at a subcellular level [8], and in the tunicate *Ciona intestinalis* to follow the fate of the central nervous system cells after metamorphosis [9]. However, the use of Kaede in non-conventional models represents a true challenge in terms of sample size, embryo immobilization and imaging. Among marine animals, cephalochordates (i.e. amphioxus) represent the most basally divergent chordate clade (also comprising vertebrates and tunicates) and the best extant proxy to the ancestor of all chordates [10]. Understanding embryonic cell trajectories in amphioxus might thus give us new insights into the evolution of vertebrate morphological novelties. Although some data, using Nile blue staining to follow the fate of blastomeres at the 4, 8, 16 and 32-cell stages, were obtained in the 60's (see [11] for a review), we are still missing a clear understanding of how the different morphological structures of the amphioxus larva develop from the two gastrula germ layers (i.e. the ectoderm and the mesendoderm). Here, we report a protocol that uses Kaede photoconversion to track cells during amphioxus embryogenesis. This protocol allows following the fate of cells from any early embryonic stage to the late neurula/larva stage and can be potentially adapted to study embryogenesis in other marine species.

## Materials and methods

The protocol described in this peer-reviewed article (Fig 1) is published on protocols.io, https://dx.doi.org/10.17504/protocols.io.j8nlk46z6g5r/v1 and is included for printing purposes as S1 File.

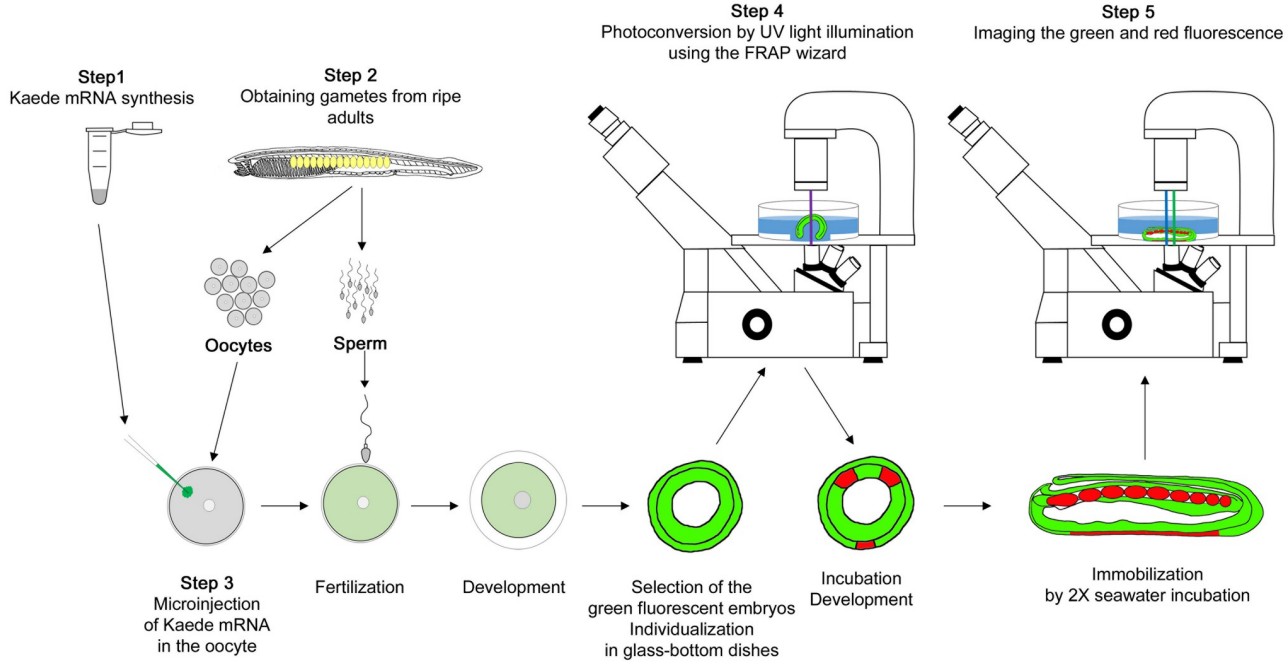

**Fig 1. Main steps for cell tracking using Kaede photoconversion in amphioxus embryos.**

## Expected results

We used the proposed protocol to follow the cells of the presumptive paraxial mesoderm territory. The photoconversion was undertaken in a gastrula stage embryo at the level of the dorsal paraxial mesendoderm on both sides as well as in a small region of the ectoderm on the opposite side of the embryo. This region corresponds to the presumptive ventral epidermis and Kaede was photoconverted in this territory in order to confirm that the photoconversion was undertaken in the correct position in the dorsal paraxial region. The embryo imaged at the late neurula stage showed expression in the ventral epidermis and in the somites (Fig 2). The same protocol was succesfully undertaken to trace the fate of the presumptive anterior somite

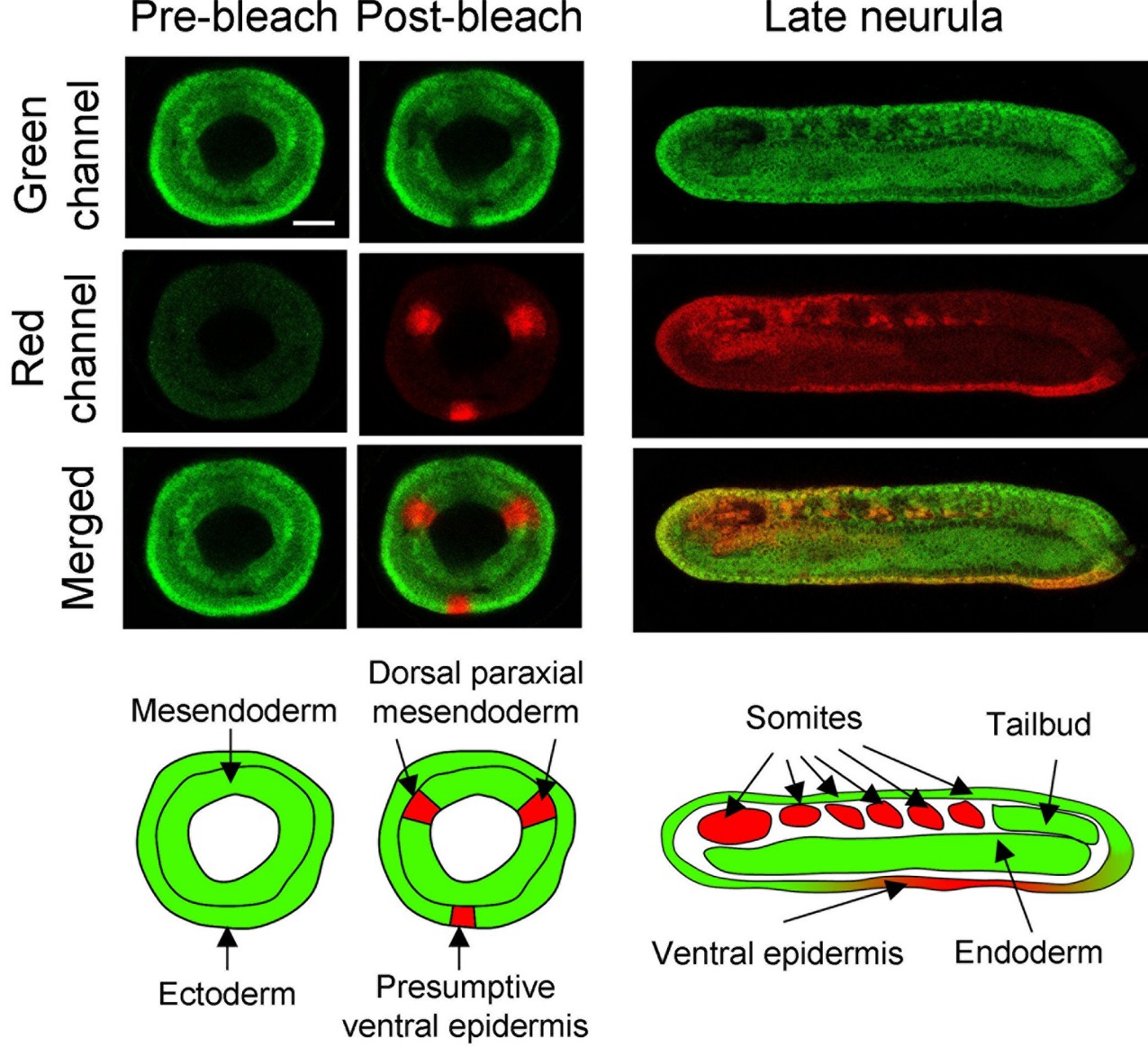

**Fig 2. Example of expected results for a cell tracking experiment using Kaede photoconversion in the amphioxus embryo.** The gastrula stage embryo (G4 stage) [13, 14] was imaged before and after photoconversion (pre- and post-bleach, panels on the left). The same embryo was imaged at the late neurula stage (N4 stage) [13, 14] (panels on the right). Gastrula stage pictures are blastopore views with dorsal to the top. Neurula stage images are side views with anterior to the left and dorsal to the top. Scale bar: 25 μm.

territory in embryos in which the FGF signalling pathway was inhibited by using a FGFR inhibitor (SU5402) as published in [12].

## Supporting information

**S1 File. Step-by-step protocol, also available on protocols.io.**
(PDF)

## Acknowledgments

This work benefited from access to the Observatoire Océanologique de Banyuls-sur-Mer, an EMBRC-France and EMBRC-ERIC site. Embryo imaging experiments were undertaken using the material of the BIOPIC platform.

## Author Contributions

**Conceptualization:** Lydvina Meister, Hector Escriva, Stéphanie Bertrand.

**Funding acquisition:** Hector Escriva.

**Investigation:** Lydvina Meister, Hector Escriva, Stéphanie Bertrand.

**Writing – original draft:** Lydvina Meister, Hector Escriva, Stéphanie Bertrand.

**Writing – review & editing:** Lydvina Meister, Hector Escriva, Stéphanie Bertrand.

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
