## [Decision Letter · Decision Letter 0]

11 Aug 2022

PONE-D-22-17287Exploring tissue morphodynamics using the photoconvertible Kaede protein in amphioxus embryosPLOS ONE

Dear Dr. Bertrand,

Thank you for submitting your manuscript to PLOS ONE. After careful consideration, we feel that it has merit but does not fully meet PLOS ONE’s publication criteria as it currently stands. Therefore, we invite you to submit a revised version of the manuscript that addresses the points raised during the review process.

As a short protocol article, this work contains useful information to the research community. One reviewer and myself have provided several comments, in the hope to further improve the clarity of the manuscript. I suggest the authors to follow those comments to revise this manuscript.

We look forward to receiving your revised manuscript.

Kind regards,

Jr-Kai Sky Yu, Ph.D.

Academic Editor

PLOS ONE

Journal Requirements:

2. We note you have not yet provided a protocols.io PDF version of your protocol and/or a protocols.io DOI. When you submit your revision, please provide a PDF version of your protocol as generated by protocols.io (the file will have the protocols.io logo in the upper right corner of the first page) as a Supporting Information file. The filename should be S1_file.pdf, and you should enter “S1 File” into the Description field. Any additional protocols should be numbered S2, S3, and so on. Please also follow the instructions for Supporting Information captions [https://journals.plos.org/plosone/s/supporting-information#loc-captions]. The title in the caption should read: “Step-by-step protocol, also available on protocols.io.”

Please assign your protocol a protocols.io DOI, if you have not already done so, and include the following line in the Materials and Methods section of your manuscript: “The protocol described in this peer-reviewed article is published on protocols.io (https://dx.doi.org/10.17504/protocols.io.[...]) and is included for printing purposes as S1 File.” You should also supply the DOI in the Protocols.io DOI field of the submission form when you submit your revision.

If you have not yet uploaded your protocol to protocols.io, you are invited to use the platform’s protocol entry service [https://www.protocols.io/we-enter-protocols] for doing so, at no charge. Through this service, the team at protocols.io will enter your protocol for you and format it in a way that takes advantage of the platform’s features. When submitting your protocol to the protocol entry service please include the customer code PLOS2022 in the Note field and indicate that your protocol is associated with a PLOS ONE Lab Protocol Submission. You should also include the title and manuscript number of your PLOS ONE submission.

"This work benefited from access to the Observatoire Océanologique de Banyuls-sur-Mer, an EMBRC-France and EMBRC-ERIC site. Embryo imaging experiments were undertaken using the material of the BIOPIC platform. The laboratory of H.E. was supported by the CNRS, by the “Agence Nationale de la Recherche” under the grants ANR-19-CE13-0011-01 and ANR-16-CE12-0008-01), and by the European project Assemble Plus (H2020-INFRAIA-1-2016- 2017; grant no. 730984). S.B. was supported by the Institut Universitaire de France".

 "HE recieved financial support from the “Agence Nationale de la Recherche” under the grants ANR-19-CE13-0011-01 and ANR-16-CE12-0008-01and from the European project Assemble Plus (H2020-INFRAIA-1-2016- 2017; grant no. 730984)."

Additional Editor Comments:

I have a few comments on the protocol:

1. File S1, page 4, section 3.2, 1st step; please check the concentration of Fast Green FCF used here (18%). It seems much higher than the listed solubility (1mg/mL) on the Sigma-Aldrich catalog.

2. File S1, page 5, section 3.2, 6th step; it should be more specific on this description of “a small volume”. To my personal knowledge, the right amount of injection volume is crucial for the successful rate of microinjection experiments, thus I would suggest the authors to provide more specific description for determining this injection volume.

Reviewers' comments:

Reviewer's Responses to Questions

**Comments to the Author**

1. Does the manuscript report a protocol which is of utility to the research community and adds value to the published literature?

Reviewer #1: Yes

Reviewer #2: No

2. Has the protocol been described in sufficient detail?

Descriptions of methods and reagents contained in the step-by-step protocol should be reported in sufficient detail for another researcher to reproduce all experiments and analyses. The protocol should describe the appropriate controls, sample sizes and replication needed to ensure that the data are robust and reproducible.

Reviewer #1: Yes

Reviewer #2: No

3. Does the protocol describe a validated method?

Reviewer #1: Yes

Reviewer #2: Yes

4. If the manuscript contains new data, have the authors made this data fully available?

Reviewer #1: N/A

Reviewer #2: N/A

**5. Is the article presented in an intelligible fashion and written in standard English?**

Reviewer #1: Yes

Reviewer #2: Yes

6. Review Comments to the Author

Reviewer #1: PONE-D-22-17287

Exploring tissue morphodynamics using the photoconvertible Kaede protein in

amphioxus embryos

The protocol is useful for a growing cohort of researchers in this animal. The animal (amphioxus) is not new to research, nor is the protein used (Kaede) but the combination here is quite useful.

“published” protocols.io is not a functional link

Demonstrable success using this method is given and adequately shows the effectiveness of the protocol and goals.

Has nice links to JOVE references for ease of learning the procedure.

Sufficient detail is given for the protocols, and are comprehensive.

Additional microscope protocols would be helpful since not everyone uses the same as described herein.

Reviewer #2: The protocol described in this peer-reviewed article (Figure 1) is published already on protocols.io. The method itself is quite straight forward by injecting mRNA of fluorescent protein.

7. PLOS authors have the option to publish the peer review history of their article (what does this mean?). If published, this will include your full peer review and any attached files.

Reviewer #1: No

Reviewer #2: No

---

## [Author Response · Author response to Decision Letter 0]

9 Sep 2022

Dear editor,

plesae find below our answers to the comments on our manuscript PONE-D-22-17287, “Exploring tissue morphodynamics using the photoconvertible Kaede protein in amphioxus embryos”. 

Additional Editor Comments:

1.File S1, page 4, section 3.2, 1st step; please check the concentration of Fast Green FCF used here (18%). It seems much higher than the listed solubility (1mg/mL) on the Sigma-Aldrich catalog.

Thank you for pointing out this error. The 18% correspond to the final volume percentage of a solution of fast green diluted in water. The initial solution is at 10mg/mL (the solubility being, contrary to what is indicated in the Sigma-Aldrich catalog, 20g/100ml.Please see: https://pubchem.ncbi.nlm.nih.gov/compound/Fast-Green-FCF#section=Solubility). We have modified the protocol on protocols.io accordingly.

2. File S1, page 5, section 3.2, 6th step; it should be more specific on this description of “a small volume”. To my personal knowledge, the right amount of injection volume is crucial for the successful rate of microinjection experiments, thus I would suggest the authors to provide more specific description for determining this injection volume.

Thank you for the remark. We have modified as follows to be more precise: “Insert the needle into the oocyte and inject a small volume of injection mix (1/100 to 1/50 of the volume of the oocyte). Depending on the size of the needle after cutting, several injection pulses might be necessary to inject a sufficient volume.”

Reviewer #1: 

Exploring tissue morphodynamics using the photoconvertible Kaede protein in amphioxus embryos

The protocol is useful for a growing cohort of researchers in this animal. The animal (amphioxus) is not new to research, nor is the protein used (Kaede) but the combination here is quite useful.

We would like to thank the reviewer for this kind comment.

“published” protocols.io is not a functional link

We have now “published” the protocol and give the correct link to the protocols.io file.

Demonstrable success using this method is given and adequately shows the effectiveness of the protocol and goals.

Has nice links to JOVE references for ease of learning the procedure.

Sufficient detail is given for the protocols, and are comprehensive.

Thanks again.

Additional microscope protocols would be helpful since not everyone uses the same as described herein.

We understand this concern. As requested by PloS ONE for Lab Protocols article, we only described the protocol that has been validated in our laboratory. However, we have added in step 4.1 the following paragraph that should allow researchers having a different microscope to consider undertaking an experiment of this type with their own equipment:

“However, any confocal inverted microscope equipped with a UV laser and lasers to image the fluorescence emitted by the Kaede protein can be used. The microscope must also allow scan zoom and ROI scanning in order to effectively target a specific region using this protocol for photoconversion. The UV laser intensity and scan time must be adjusted. If a FRAP module or a photoconversion/photoactivation module is available on the microscope, it can be used following the manufacturer's instructions.”.

Reviewer #2:

The protocol described in this peer-reviewed article (Figure 1) is published already on protocols.io. The method itself is quite straight forward by injecting mRNA of fluorescent protein.

We are sorry we don’t understand this remark. There is no such protocol in protocols.io (nor on amphioxus injection, nor on Kaede photoconversion).

---

## [Editor Report · Decision Letter 1]

12 Sep 2022

Exploring tissue morphodynamics using the photoconvertible Kaede protein in amphioxus embryos

PONE-D-22-17287R1

Dear Dr. Bertrand,

We’re pleased to inform you that your manuscript has been judged scientifically suitable for publication and will be formally accepted for publication once it meets all outstanding technical requirements.

Kind regards,

Jr-Kai Yu, Ph.D.

Academic Editor

PLOS ONE
---

## [Editor Report · Acceptance letter]

19 Sep 2022

PONE-D-22-17287R1 

Exploring tissue morphodynamics using the photoconvertible Kaede protein in amphioxus embryos 

Dear Dr. Bertrand:

I'm pleased to inform you that your manuscript has been deemed suitable for publication in PLOS ONE. Congratulations! Your manuscript is now with our production department. 

Kind regards, 

on behalf of

Dr. Jr-Kai Yu 

Academic Editor

PLOS ONE